# Leveraging Clinical Characteristics for Improved Deep Learning-Based Kidney Tumor Segmentation on CT

Christina B. Lund and Bas H.M. van der Velden

Image Sciences Institute, UMC Utrecht, Utrecht University, Utrecht, The Netherlands
C.B.Lund-2@umcutrecht.nl
B.H.M.vanderVelden-2@umcutrecht.nl

**Abstract.** This paper assesses whether using clinical characteristics in addition to imaging can improve automated segmentation of kidney cancer on contrast-enhanced computed tomography (CT). A total of 300 kidney cancer patients with contrast-enhanced CT scans and clinical characteristics were included. A baseline segmentation of the kidney cancer was performed using a 3D U-Net. Input to the U-Net were the contrast-enhanced CT images, output were segmentations of kidney, kidney tumors, and kidney cysts. A cognizant sampling strategy was used to leverage clinical characteristics for improved segmentation. To this end, a Least Absolute Shrinkage and Selection Operator (LASSO) was used. Segmentations were evaluated using Dice and Surface Dice. Improvement in segmentation was assessed using Wilcoxon signed rank test. The baseline 3D U-Net showed a segmentation performance of 0.90 for kidney and kidney masses, i.e., kidney, tumor, and cyst, 0.29 for kidney masses, and 0.28 for kidney tumor, while the 3D U-Net trained with cognizant sampling enhanced the segmentation performance and reached Dice scores of 0.90, 0.39, and 0.38 respectively. To conclude, the cognizant sampling strategy leveraging the clinical characteristics significantly improved kidney cancer segmentation.

**Keywords:** Kidney Cancer · Deep Learning · Cognizant Sampling · Clinical Characteristics · Automated Semantic Segmentation

## 1 Introduction

According to World Health Organization a total of 431,288 people were diagnosed with kidney cancer in 2020. This makes kidney cancer the 14th most common cancer worldwide [1]. Although the number of new cases is relatively high, many patients present asymptomatic until the cancer has metastasized, and more than fifty percent of all cases are thus discovered incidentally on abdominal imaging examinations performed for other purposes [2]. Masses suspected of malignancy are investigated predominantly with contrast-enhanced computed tomography (CT) or magnetic resonance imaging (MRI) [2]. Information about size, location,

and morphology of the tumor can enhance treatment decisions, but manual evaluation of the CT scans remains laborious work. Scoring systems such as the R.E.N.A.L Nephrometry Score and PADUA exist to steer manual evaluation [3, 4], but are subject to interobserver variability [5].

Treatment of localized kidney cancer consists of surgery of tumor and immediate surroundings (i.e., partial nephrectomy), surgery of tumor and entire kidney (i.e., radical nephrectomy), or active surveillance in case of patients who do not undergo surgery immediately but are carefully followed and evaluated for signs of disease progression [2].

Computer decision-support systems have potential to personalize treatment. Examples of such systems include volumetric measurements and radiomics approaches [6]. A crucial first step in these systems is to accurately identify kidney and kidney cancer.

The 2021 Kidney and Kidney Tumor Segmentation Challenge (KiTS21) provides a platform for researchers to test software dedicated to segmenting kidney cancer. KiTS21 does not only include images and corresponding annotations, but also an extensive set of clinical characteristics. The organizers of KiTS19 investigated whether imaging and clinical characteristics affected segmentation performance and found that tumor size had a significant association with tumor Dice score [7]. Therefore, it is reasonable to assume there may be other clinical characteristics that can be leveraged to improve segmentation.

The aim of our study was to assess whether using clinical characteristics in addition to imaging can improve the segmentation of kidney cancer.

## 2   Material and Methods

We compared two different strategies (Figure 1). As baseline, we used a 3D U-Net. We propose to improve this baseline by investigating in the validation set which clinical characteristics affect the model's performance and leverage this for cognizant sampling.

### 2.1   Training and Validation Data

Our submission exclusively used data from the official KiTS21 training set. The dataset contained contrast-enhanced preoperative CT scans of 300 patients who underwent partial or radical nephrectomy between 2010 and 2020. Each CT scan was independently annotated by three annotators for each of the three semantic classes: Kidney, Tumor, and Cyst. To create plausible complete annotations for use during evaluation, the challenge organizers generate groups of sampled annotations. Across these groups, none of the samples have overlapping instance annotations. It is therefore possible to compare and average them without underestimating the interobserver disagreement.

We randomly divided the dataset into training, validation, and test sets, consisting of 210, 60, and 30 patients respectively.

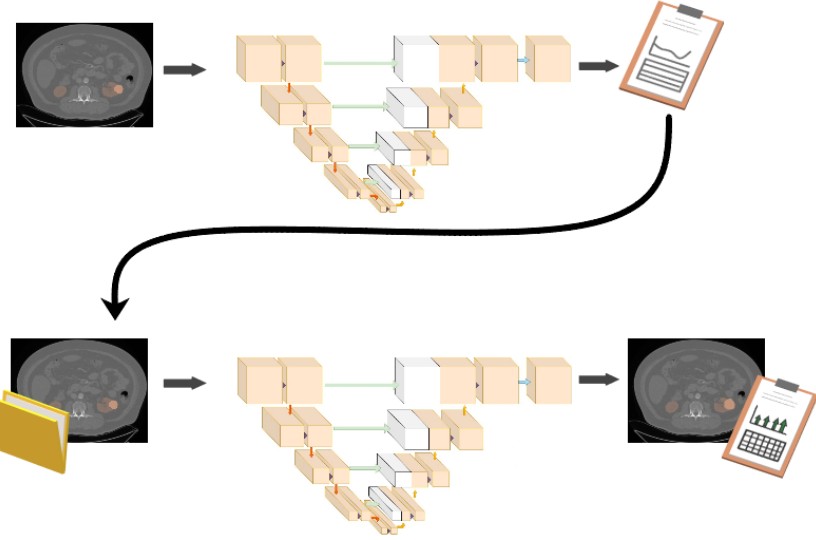

**Fig. 1.** By leveraging the performance of our baseline 3D U-Net model on the validation set (top row), we propose a cognizant sampling strategy based on clinical characteristics for improved segmentation

### 2.2   Preprocessing

Images in the dataset were acquired from more than 50 referring medical centers, leading to various acquisition protocols and thus notable differences in the image resolutions. The in-plane resolution ranged from 0.44 mm to 1.04 mm while the slice thickness ranged from 0.5 mm to 5.0 mm. To alleviate these differences, we chose to resample all images to a common resolution of 3 mm x 1.56 mm x 1.56 mm, which is the median slice thickness and twice the median in-plane resolution. Images were resampled using Lanczos interpolation, annotations were resampled using nearest neighbor interpolation. Resampling yielded a median image size of 138 x 256 x 256 voxels.

We truncated the image intensities to the 0.5 to 99.5 percentiles of the intensities of the annotated voxels in the training set. Afterwards, we performed zero-mean-unit-variance standardization based on these voxels.

Augmentations included adjustments of gamma, contrast, and brightness, addition of Gaussian noise, Gaussian blurring, scaling, rotation, and mirroring. To accommodate GPU memory limitations we cropped the images into patches of 96 x 160 x 160 voxels.

### 2.3   Baseline 3D U-Net

**Training** The baseline 3D U-Net consisted of a downsampling path followed by an upsampling path. Downsampling was performed by max pooling operations while upsampling was done with transposed convolutions. The different parameters are described in Table 1. Because of the substantial class imbalance, we defined the loss function as the equally weighted sum of the Dice and a weighted Cross Entropy. We used Adam as optimizer with an initial learning rate of 0.005. The learning rate was reduced with a factor 0.3 if there had been no improvement in the validation loss during the last 10 epochs. The model was trained from scratch for 100 epochs. Each epoch included 400 volumes randomly sampled from the training set in batches of two. All deep learning was performed using PyTorch version 1.5.0.

**Evaluation** During inference we used a sliding window (size 96 x 160 x 160 voxels) to cover the entire volume. Windows overlapped with half the window size. Postprocessing consisted of retaining the two largest connected components using anatomical prior knowledge. We evaluated the model's predictions using the KiTS21 evaluation script with sampled annotations as the ground truth (see section 2.1). This evaluation uses three hierarchical evaluation classes: Kidney and Masses (Kidney + Tumor + Cyst), Masses (Tumor + Cyst), and Tumor in combination with six evaluation metrics: Dice and Surface Dice scores of the three classes [8].

### 2.4   Cognizant Sampling Leveraging Clinical Characteristics

To devise a cognizant sampling strategy, we investigated the effect of clinical characteristics on the model's performance on the validation set. We investigated all clinical characteristics that had complete cases (i.e., no missing data) and had contrast between the patients (i.e., the variable was not the same value for all patients).

The Least Absolute Shrinkage and Selection Operator (LASSO) was used to assess which characteristics were significantly associated with kidney tumor Dice. The LASSO uses L1 regularization, which has the advantage that a sparse subset of characteristics was selected [9]. Clinical characteristics were normalized before LASSO analysis, LASSO used 5-fold cross validation.

The characteristics associated with kidney tumor Dice were weighted by the inverse of the frequency of those characteristics in the cognizant sampling strategy. For example, if smoking history was associated with kidney tumor Dice and 50% of the patients in the training population smoked, the weights of the non-smoker subset was set twice as large during cognizant sampling.

The model was retrained with no other changes than the application of the cognizant sampling strategy.

**Table 1.** Network description

| Layer name | Layer description | Output dimension |
|---|---|---|
| Input | Input | 1 x 96 x 160 x 160 |
| Dconv1 | Double convolution block: 2x (3D Convolution - Instance Normalization - ReLU Activation) Convolution kernel size: 3x3x3, stride: 1x1x1, padding: 1 | 24 x 96 x 160 x 160 |
| Mpool | Downsampling, level 1 Max pooling kernel size: 2x2x2, stride: 2x2x2 | 24 x 48 x 80 x 80 |
| Dconv2 | Double convolution block | 48 x 48 x 80 x 80 |
| Mpool | Downsampling, level 2 | 48 x 24 x 40 x 40 |
| Dconv3 | Double convolution block | 96 x 24 x 40 x 40 |
| Mpool | Downsampling, level 3 | 96 x 12 x 20 x 20 |
| Dconv4 | Double convolution block | 192 x 12 x 20 x 20 |
| Mpool | Downsampling, level 4 | 192 x 6 x 10 x 10 |
| Dconv5 | Double convolution block | 384 x 6 x 10 x 10 |
| Tconv4 | Upsampling, level 4 Transposed Convolution kernel size: 2x2x2, stride: 2x2x2 | 192 x 12 x 20 x 20 |
| Concat | Concatenation: [Dconv4, Tconv4] | 384 x 12 x 20 x 20 |
| Dconv6 | Double convolution block | 192 x 12 x 20 x 20 |
| Tconv3 | Upsampling, level 3 | 96 x 24 x 40 x 40 |
| Concat | Concatenation: [Dconv3, Tconv3] | 192 x 24 x 40 x 40 |
| Dconv7 | Double convolution block | 96 x 24 x 40 x 40 |
| Tconv2 | Upsampling, level 2 | 48 x 48 x 80 x 80 |
| Concat | Concatenation: [Dconv2, Tconv2] | 96 x 48 x 80 x 80 |
| Dconv8 | Double convolution block | 48 x 48 x 40 x 40 |
| Tconv1 | Upsampling, level 1 | 24 x 96 x 160 x 160 |
| Concat | Concatenation: [Dconv1, Tconv1] | 48 x 96 x 160 x 160 |
| Dconv9 | Double convolution block | 24 x 96 x 160 x 160 |
| Output | 3D Convolution and Softmax activation Convolution kernel size: 1x1x1, stride: 1x1x1, padding: 0 | 4 x 96 x 160 x 160 |

## 2.5   Statistical Evaluation

We evaluated segmentation performance (i.e., (Surface) Dice scores) of the baseline model and the model with the cognizant sampling on the test set (N = 30 patients). Normality of these performance scores was assessed using the Shapiro-Wilk test. Statistical differences in performance were assessed using the paired t-test in case of normal distributions and using the Wilcoxon signed ranked test in case of non-normal distributions. A P-value below 0.05 was considered statistically significant. All statistical analyses were performed using R version 3.6.1.

## 3    Results

Output of the LASSO showed that presence of chronic kidney disease, a history of smoking, larger tumor size, and radical nephrectomy instead of partial nephrectomy yielded higher tumor Dice scores (Table 2, Figure 2).

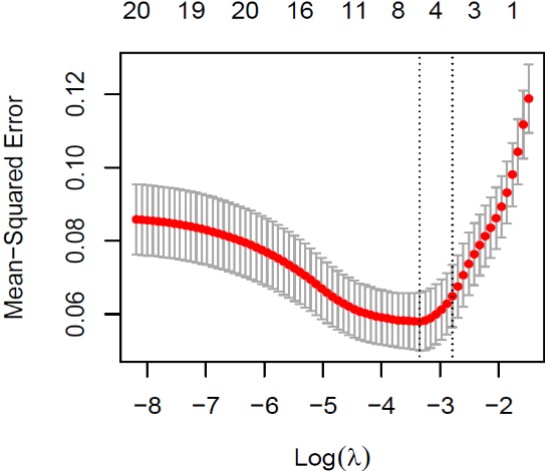

**Fig. 2.** Least Absolute Shrinkage and Selection Operator (LASSO) analysis shows that four variables are associated with kidney tumor Dice at one standard error from the minimum.

**Table 2.** The four variables associated with kidney tumor Dice in the validation set according to the Least Absolute Shrinkage and Selection Operator (LASSO)

| Variable | Coefficient |
| --- | --- |
| Intercept | 0.108 |
| Comorbidities: Chronic kidney disease | 0.118 |
| Smoking history: Previous smoker | 0.076 |
| Radiographic size | 0.065 |
| Surgical procedure: Radical nephrectomy | 0.050 |

The cognizant sampling strategy significantly improved the model's segmentation performance (Table 3, Figure 3).

**Table 3.** Dice and Surface Dice scores for the model trained with random sampling and the model trained with the proposed cognizant sampling strategy. SD = standard deviation, PR = percentile range.

| Dice | Kidney | Masses | Tumor |
| --- | --- | --- | --- |
| **Random Sampling** | | | |
| Mean (SD) | 0.90 (0.08) | 0.29 (0.28) | 0.28 (0.29) |
| Median (25-75 PR) | 0.93 (0.88-0.94) | 0.17 (0.03-0.55) | 0.15 (0.03-0.57) |
| **Cognizant Sampling** | | | |
| Mean (SD) | 0.90 (0.12) | 0.39 (0.32) | 0.38 (0.34) |
| Median (25-75 PR) | 0.95 (0.92-0.95) | 0.42 (0.07-0.70) | 0.37 (0.03-0.71) |
| **P-value** | 0.004 | 0.013 | 0.033 |

| Surface Dice | Kidney | Masses | Tumor |
| --- | --- | --- | --- |
| **Random Sampling** | | | |
| Mean (SD) | 0.74 (0.14) | 0.12 (0.11) | 0.11 (0.11) |
| Median (25-75 PR) | 0.79 (0.66-0.84) | 0.07 (0.03-0.18) | 0.06 (0.03-0.18) |
| **Cognizant Sampling** | | | |
| Mean (SD) | 0.78 (0.16) | 0.23 (0.18) | 0.23 (0.21) |
| Median (25-75 PR) | 0.84 (0.73-0.90) | 0.19 (0.05-0.38) | 0.22 (0.01-0.39) |
| **P-value** | 0.003 | <0.001 | <0.001 |

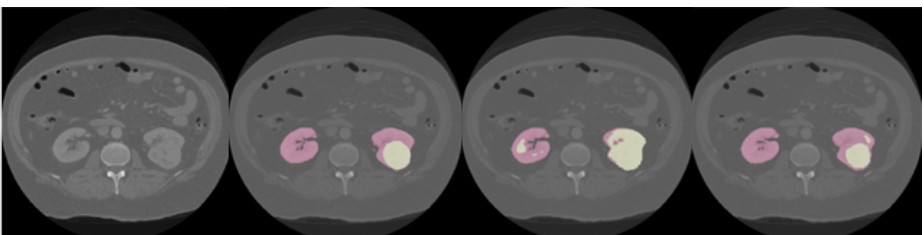

**Fig. 3.** Example of a 66 year old patient from the test set in whom the cognizant sampling leveraging clinical characteristics improved segmentation of the cancer. From left to right: Image, ground truth annotations, segmentation from baseline 3D U-Net, segmentation from the model using cognizant sampling

## 4  Discussion and Conclusion

A cognizant sampling strategy leveraging clinical characteristics significantly improved segmentation of kidney cancer on contrast-enhanced CT.

A baseline 3D U-Net was trained using random sampling. The clinical characteristics that were most associated with segmentation performance were identified using LASSO regression and used in a cognizant sampling strategy thereby leveraging the effect of the identified clinical characteristics. Previous studies showed that data-driven weighting can yield results that are independent of clin-

ical characteristics [10, 11]. Such approaches have the potential to eliminate bias towards characteristics such as smoking history, but potentially also undesirable bias such as in gender or race.

Our baseline model was a standard 3D U-Net instead of e.g. the nnU-Net provided by the challenge organizers. Since the aim of our study was to assess whether using clinical characteristics in addition to imaging can improve the segmentation of kidney cancer, we chose to direct our attention towards leveraging the potential effect of the clinical data rather than focusing solely on outperforming the results obtained with nnU-Net. It is plausible that leveraging the clinical characteristics could also improve other network architectures such as nnU-Net, because it can circumvent potential biases in patient populations. To conclude, we showed that cognizant sampling leveraging clinical characteristics improves segmentation of kidney cancer on contrast-enhanced CT.

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
