# OpenReview forum: "Leveraging Clinical Characteristics for Improved Deep Learning-Based Kidney Tumor Segmentation on CT"
_MICCAI.org/2021/Challenge/KiTS — Submitted to KiTS21 Challenge_

### Official Review · Reviewer_RPT2 · 2021-08-30

**Rating:** 8

**Review:**

The authors present a very interesting study in which they leveraged each case's associated clinical data to stratify cases in terms of their expected performance, and then (I believe) they oversampled those cases during training later on, in kind of a "hard example mining" approach. Their experiments show that this method improved performance over their baseline. The paper is well written and a good length. It would be nice if they could explain in more detail what "cognizant sampling" actually means -- how exactly did the clinical information change your training strategy?

---

### Official Review · Reviewer_czcj · 2021-08-30

**Rating:** 10

**Review:**

### Overall

- I know that clinical abstracts often have headers in them, but I don't think that springer will allow us to include them. Please combine the abstract into a single block of text without headers.

### Introduction

- Looks good

### Methods

- What strategy did you use for resampling/interpolation?

### Results

- Very nice presentation of results thus far. Please make sure to also add the final results once they are known

### Discussion and Conclusion

- No comments

---

### Decision · Program_Chairs · 2021-08-30

**Decision:**

Minor Revisions

**Comment:**

Please address the reviewer comments and resubmit